# The seroincidence of childhood *Shigella sonnei* infection in Ho Chi Minh City, Vietnam

**Nick K. Jones**[1]☯*, **Trang Nguyen Hoang Thu**[2]☯, **Ruklanthi de Alwis**[2,3], **Corinne Thompson**[2], **Ha Thanh Tuyen**[2], **Tran Do Hoang Nhu**[2], **Voong Vinh Phat**[2], **Pham Duc Trung**[2], **Phung Khanh Lam**[2], **Bui Thi Thuy Tien**[4], **Hoang Thi Diem Tuyet**[4], **Lu Lan Vi**[5], **Nguyen Van Vinh Chau**[5], **Nhi Le Thi Quynh**[2], **Stephen Baker**[1,2]*

**1** Department of Medicine, University of Cambridge, Cambridge, United Kingdom, **2** Oxford University Clinical Research Unit, Ho Chi Minh City, Vietnam, **3** Programme in Emerging Infectious Diseases, Duke-NUS Medical School, Singapore, Singapore, **4** Hung Vuong Hospital, Hong Bang, Ho Chi Minh City, Vietnam, **5** The Hospital for Tropical Diseases, Vo Van Kiet, Ho Chi Minh City, Vietnam

☯ These authors contributed equally to this work.
* nicholas.jones20@nhs.net (NKJ); sgb47@cam.ac.uk (SB)

## Abstract

### Background

*Shigella sonnei* is a pathogen of growing global importance as a cause of diarrhoeal illness in childhood, particularly in transitional low-middle income countries (LMICs). Here, we sought to determine the incidence of childhood exposure to *S. sonnei* infection in a contemporary transitional LMIC population, where it represents the dominant *Shigella* species.

### Methods

Participants were enrolled between the age of 12–36 months between June and December 2014. Baseline characteristics were obtained through standardized electronic questionnaires, and serum samples were collected at 6-month intervals over two years of follow-up. IgG antibody against *S. sonnei* O-antigen (anti-O) was measured using an enzyme-linked immunosorbent assay (ELISA). A four-fold increase in ELISA units (EU) with convalescent IgG titre >10.3 EU was taken as evidence of seroconversion between timepoints.

### Results

A total of 3,498 serum samples were collected from 748 participants; 3,170 from the 634 participants that completed follow-up. Measures of anti-O IgG varied significantly by calendar month ($p = 0.03$). Estimated *S. sonnei* seroincidence was 21,451 infections per 100,000 population per year (95% CI 19,307–23,834), with peak incidence occurring at 12–18 months of age. Three baseline factors were independently associated with the likelihood of seroconversion; ever having breastfed (aOR 2.54, CI 1.22–5.26), history of prior hospital admission (aOR 0.57, CI 0.34–0.95), and use of a toilet spray-wash in the household (aOR 0.42, CI 0.20–0.89).

**Data Availability Statement:** All relevant data are presented within the manuscript and its supporting files.

**Funding:** This work was supported by a Wellcome senior research fellowship to SB (215515/Z/19/Z). The funders had no role in study design, data collection and analysis, decision to publish, or preparation of the manuscript.

**Competing interests:** The authors have declared that no competing interests exist.

## Conclusions

Incidence of *S. sonnei* exposure in Ho Chi Minh City is substantial, with significant reduction in the likelihood of exposure as age increases beyond 2 years.

## Author summary

Bacteria belonging to the genus *Shigella* are a leading cause of childhood diarrhea globally. In Vietnam and other transitional low-middle income countries (LMIC), *S. sonnei* has recently emerged as the dominant *Shigella* species, displacing *S. flexneri*, which usually predominates in industrializing regions. Little is known about the epidemiology of *S. sonnei* and how common infection is amongst children in such settings. In this study, we used regular blood sampling from a large cohort of children in Ho Chi Minh City over a two year period to examine changes in antibody concentrations against *S. sonnei*, using this to detect evidence of exposure to the bacteria. Importantly, this approach allowed us to identify exposures irrespective of whether the children had experienced symptoms or not. Our findings showed that exposure to *S. sonnei* was common in this cohort, with the highest incidence of exposure occurring in children observed between the ages of 12 months and 18 months. This is the first study of its kind to describe the epidemiology of *S. sonnei* infection in a LMIC setting and will be useful in informing future vaccine development and deployment strategies against this important pathogen.

## Introduction

Bacteria belonging to the genus *Shigella* cause shigellosis, a severe form of diarrhoea characterized by blood and mucus in the stool [1]. There are estimated to be >180 million cases of shigellosis per year worldwide, of which >1,000,000 are fatal [2]. The preponderance of shigellosis cases occurs in young children (<5 years) in low-middle income countries (LMICs), giving rise to increased risk of mortality and developmental complications [3].

The *Shigella* genus is comprised of four species: *S. sonnei*, *S. flexneri*, *S. boydii*, and *S. dysentariae*. *S. flexneri* and *S. sonnei* predominate globally, with *S. flexneri* classically associated with industrializing countries and *S. sonnei* more commonly isolated in industrialized regions. However, *S. sonnei* is an emergent pathogen, and has almost completely replaced *S. flexneri* as the most common agent of shigellosis in some rapidly developing LMICs [4]. In a recent study of >3,000 children hospitalized with severe diarrhea in Ho Chi Minh City, Vietnam, 99% of all isolated *Shigella* spp. were *S. sonnei* [5]. The rapid dominance of *S. sonnei* is compounded by its propensity to develop multi-drug resistance (MDR); the recent emergence of extended-drug resistance (XDR; MDR plus resistance to fluoroquinolones, third generation cephalosporins, macrolides, and aminoglycosides) makes treatment and disease control efforts with antibiotics increasingly futile [6].

Epidemiological studies that specifically focus on *S. sonnei* are limited, as the organism has been largely overlooked in favour of *S. flexneri* and is assumed to follow comparable disease mechanisms. However, despite being of the same genus, *S. sonnei* and *S. flexneri* are otherwise unrelated, having arisen via convergent evolution to cause clinically indistinguishable infections [1]. Disparities in their evolutionary histories, pathogenesis, fluctuations in global distribution, potential transmission routes, and interaction with antibiotics suggest that they should be considered epidemiologically distinct [4]. Indeed, whilst vaccines targeting the dominant *S.*

*flexneri* serotypes have been a major focus of international research in recent years [7], the neglect of *S. sonnei* as an independent contributor to the global burden of childhood dysentery and antimicrobial resistance risks hampering efforts to reduce prevalence through the development and deployment of vaccines, particularly in transitional LMICs.

*S. sonnei*, unlike other *Shigella* spp., is characterized by a single serotype, which is determined by the repeating O-antigen component of the surface lipopolysaccharide (LPS). *S. sonnei*-O antigen is the principal antigen observed by the immune system during infection, resulting in anti-*S. sonnei*-O IgG being an accepted marker of *S. sonnei* infection/exposure. Although not a definitive correlate of protective immunity (except after experimental vaccines) [8], anti-*S. sonnei*-O IgG is a key indicator of acquired immunity, with a lack of antibody associated with an increased risk of symptomatic disease [9]. Notably, titres of anti-*S. sonnei*-O IgG rise significantly after symptomatic infection, peak two weeks post infection (>4-fold increase) and drop to approximately two-fold higher than baseline eight weeks later [8].

Previous work has determined that whilst anti-*S. sonnei*-O IgG is efficiently transferred across the placenta, it declines rapidly and is undetectable in the majority of children at five months of age, leaving them vulnerable to infection [10]. An additional study showed that anti-*Shigella* LPS IgG rises dramatically after birth, peaks at 3–4 years of age and then plateaus [8]. Exposure to *S. sonnei* has not yet been systematically measured in a contemporary transitional LMIC population, where *S. sonnei* is the predominant *Shigella* species [11]. Here, hypothesizing substantial *S. sonnei* exposure via asymptomatic infection, and aiming to inform the progression of *S. sonnei* (or polyvalent *Shigella*) vaccines by providing insight into circulating levels of naturally occurring anti-O IgG, we sought to measure the seroincidence of *S. sonnei* in a longitudinal cohort of 748 children aged <5 years residing in Ho Chi Minh City, Vietnam.

## Methods

### Ethics statement

Ethical approval for this study was provided by the Oxford tropical research ethics committee (OxTREC) in the United Kingdom (approval 1058–13); local ethical approvals were provided by the ethics committee of the Hospital for Tropical Diseases and Hung Vuong Hospital in Ho Chi Minh City, Vietnam. Written informed consent was obtained from the parents/guardians of children at the time of enrolment.

### Study design and participants

This was a prospective, community-based longitudinal study of diarrhoeal disease. Participants were sought from a prior birth cohort, the design of which has been described previously [12,13]. No additional exclusion criteria were applied during recruitment to the present study [14]. In total, 748 children aged between 12 and 36 months at the time of recruitment were enrolled into this cohort between June and December 2014, and were followed for two years (all follow ups were completed by December 2016). Study nurses approached parents to invite their children to join the cohort when they visited Hung Vuong Hospital for their last routine visit for the existing birth cohort study, or were recalled via telephone. All participants were residents of district 8 in Ho Chi Minh City and were requested to attend routine visits to the study clinic at Hung Vuong Hospital at six monthly intervals. Children who moved out of the study area for more than six months were excluded from further participation.

Study nurses used a standardized electronic questionnaire to gather information regarding socioeconomics, demographics, water and food usage, and personal hygiene practices at baseline. History of health during the preceding six months, and the height and weight of each

child was also recorded. During each routine visit venous blood samples were collected in EDTA tubes (2 ml from children aged 12–35 months and 3 ml for children aged ≥36 months), which were then separated into cells and plasma for storage at -20˚C until required.

## Laboratory methods

IgG antibody against *S. sonnei* O-antigen was measured in the plasma samples using an enzyme-linked immunosorbent assay (ELISA). Anti-O IgG was selected because of a known specificity to detect *S. sonnei* seroconversion and limited cross reactivity with other species of enterobacteriaceae, aside from the waterborne organism, *Plesiomonas shigelloides* [15]. Purified *S. sonnei* O-antigen was extracted as previously described [16] and provided by Sclavo Behring Vaccines Institute for Global Health (Siena, Italy). For the ELISA assays, 96-well microtiter plates (Maxisorb; NUNC) were coated overnight with 0.5mg/mL *S. sonnei* O-antigen in PBS pH 7.0 at 4˚C. Plates were then washed and blocked in PBS containing 5% skimmed milk for 2 hours. After washing, 100μl of each plasma sample (diluted at 1:200 in PBS containing 1% skimmed milk) were added and plates were incubated for 2 hours at room temperature. IgG against *S. sonnei* O-antigen was detected by incubation with alkaline phosphatase directly conjugated anti-human IgG for 1 hour. Plates were developed by p-nitrophenyl-phosphate solution (Sigma) and were read at absorbance 405nm and 490nm by an ELISA platereader (Microplate reader, Biorad). Each plate contained a 2-fold serially diluted pool of anti- *S. sonnei*-O antigen human plasma (primary concentration 1:200). A standard curve was generated from the corresponding optical density (OD) and ELISA units using a 4-parameter logistic regression fit. One ELISA unit (EU) was defined as the reciprocal dilution of the standard plasma that gave an absorbance value equal to 1 in this assay. The ELISAs were performed in duplicate. IgG units in the plasma samples were calculated relative to this standard each time the assay was performed.

## Outcome definitions

Seronegative was defined as anti-O IgG titre ≤10.3 EU. This threshold was based upon the median IgG level observed prior to seroconversion in Thompson *et al.* 2016 [10]. Seroconversion to seropositive was defined as a four-fold increase in EU, with convalescent IgG titre >10.3 EU.

## Statistical analyses

Pearson's chi-squared test was used to compare rates of seroconversion events by sex, age, and calendar month of sampling. The K-sample equality-of-medians test was used to compare anti-O IgG measures by sex, age and calendar month of sampling.

Factors associated with anti-O IgG levels were analysed with univariable and multivariable linear regression, adjusting for sex, age at the time of sampling, calendar month at the time of sampling, and calendar month of ELISA testing. Baseline factors associated with seroconversion over 2 years of follow-up were analysed with univariable and multivariable logistic regression. Variables for inclusion in the multivariable model were selected using a backwards elimination method based on evidence of significant association in the univariable logistic regression analysis (p<0.05). Age and calendar month of baseline serum sampling were also included in the multivariable logistic regression model because of previously observed variation in incidence of shigellosis by age and season [17–20]. All analyses were performed in Stata 15 (StataCorp, College Station, TX, USA).

## Results

### Baseline characteristics

Between 27th January 2014 and 16th December 2016, a total of 748 participants were recruited to the described cohort study. Of the 748 enrolled participants, 634 (84.8%) attended all five study visits. The baseline characteristics of the 634 participants that completed follow-up are described in Table 1. Additionally, a further 85/748 (11.4%) participants attended at least two consecutive visits but did not complete all five of the scheduled visits. The baseline characteristics of the 114 participants with incomplete follow-up are described in Table A in S1 Text.

During the course of the study, 3,498 serum samples were collected; 3,170 from participants who completed all scheduled visits and 328 from those with incomplete follow-up. Fig 1 shows the distribution of ages at the time of follow-up, and consequently the ages at which anti-*S. sonnei* O-antigen (anti-O) IgG was longitudinally measured in serum. We determined that a rise in anti-O IgG in two consecutive serum samples taken six months apart, which signifies seroconversion, occurred on 290 occasions; 272 seroconversion events occurred in participants that completed all follow-up visits and 18 seroconversion events occurred in participants with incomplete follow-up. We identified evidence of two independent seroconversion events in 18 participants, all of whom had completed follow-up (S1 Fig). A further five participants had two or more consecutive four-fold rises in anti-O IgG that were not separated by seroreversion (determined by a return to anti-O IgG <10.3 EU). Ultimately, we inferred these events to be single anti-O IgG seroconversions, as opposed to multiple sequential seroconversions (S2 Fig).

### Factors associated with anti-S. sonnei O-antigen IgG titres

We found that median anti-O IgG measures did not differ significantly between sexes ($p = 0.684$ for all samples; $p = 0.814$ for samples at the point of observed seroconversion), nor did the proportion of convalescent samples from which seroconversion was observed ($p = 0.955$) (S3 Fig and Table B in S1 Text). The median anti-O IgG measurements increased significantly with age ($p<0.001$), but there was no significant difference in median anti-O IgG measures at the point of observed seroconversion ($p = 0.359$), despite the proportion of convalescent samples indicating significantly decreasing seroconversion with age ($p = 0.006$) (S4 Fig and Table C in S1 Text). Notably, we observed significant variation in the median anti-O IgG titres by calendar month at the time of sampling ($p<0.001$ for all samples; $p<0.001$ for convalescent samples at the point of observed seroconversion, with higher levels from April to August), and the number of seroconversion events varied significantly by calendar month at the time of convalescent sampling (higher December to February; $p<0.001$) (Fig 2 and Table D in S1 Text).

### The seroincidence of S. sonnei exposure

The estimated seroincidence of *S. sonnei* in the entire cohort was 21,451 infections per 100,000 population per year (95% CI; 19,307–23,834). Stratification by age at the time of sampling determined that the estimated seroincidence of *S. sonnei* per 100,000 population per year was significantly higher for children between 12 and 18 months of age (37,374; 95% CI 28,962–48,229) compared to all other six-month periods of observation over the age of 24 months; 21,134 for 24–30 months (95% CI 16,104–27,736), 20,039 for 30–36 months (95% CI 15,678–25,613), 20,183 for 36–42 months (95% CI 15,501–26,281), 21,134 for 42–48 months (95% CI 16,104–27,736), 18,699 for 48–54 months (95% CI 12,936–27,031), and 12,800 for 54–60 months (95% CI 6,702–24,448) (Fig 3 and Table G in S1 Text). Comparison with the estimated

**Table 1. Baseline characteristics of the overall cohort of participants with completed follow-up, and comparison by observed seroconversion status.**

| | | Overall (n = 634) | Seroconverters (n = 254) | Non-seroconverters (n = 380) |
|---|---|---|---|---|
| Sex | Male | 336 (53.0%) | 139 (54.7%) | 197 (51.8%) |
| | Female | 298 (47.0%) | 115 (45.3%) | 183 (48.2%) |
| Age at enrolment (months) | 12 | 198 (31.2%) | 80 (31.5%) | 118 (31.1%) |
| | 18 | 48 (7.6%) | 20 (7.9%) | 28 (7.4%) |
| | 24 | 142 (22.4%) | 62 (24.4%) | 80 (21.1%) |
| | 30 | 121 (19.1%) | 51 (20.1%) | 70 (18.4%) |
| | 36 | 125 (19.7%) | 41 (16.1%) | 84 (22.1%) |
| Calendar month of baseline sampling | January | 1 (0.1%) | 0 (0.0%) | 1 (0.3%) |
| | February | 1 (0.1%) | 1 (0.4%) | 0 (0.0%) |
| | March | 0 (0.0%) | 0 (0.0%) | 0 (0.0%) |
| | April | 0 (0.0%) | 0 (0.0%) | 0 (0.0%) |
| | May | 0 (0.0%) | 0 (0.0%) | 0 (0.0%) |
| | June | 46 (6.2%) | 15 (5.9%) | 24 (6.3%) |
| | July | 160 (21.4%) | 62 (24.4%) | 67 (17.6%) |
| | August | 114 (15.2%) | 51 (20.1%) | 43 (11.3%) |
| | September | 131 (17.5%) | 30 (11.8%) | 86 (22.6%) |
| | October | 130 (17.4%) | 42 (16.5%) | 75 (19.7%) |
| | November | 96 (12.8%) | 25 (9.8%) | 56 (14.7%) |
| | December | 69 (9.2%) | 28 (11.0%) | 28 (7.4%) |
| Obesity | Yes | 24 (3.8%) | 14 (5.5%) | 10 (2.6%) |
| | No | 350 (55.2%) | 143 (56.3%) | 207 (54.5%) |
| | Missing | 260 (41.0%) | 97 (38.2%) | 163 (42.9%) |
| Malnourishment | Yes | 6 (1.0%) | 1 (0.4%) | 5 (1.3%) |
| | No | 366 (57.7%) | 154 (60.6%) | 212 (55.8%) |
| | Missing | 262 (41.3%) | 99 (39.0%) | 163 (42.9%) |
| History of past or present breast feeding | Yes | 589 (92.9%) | 244 (96.1%) | 345 (90.8%) |
| | No | 45 (7.1%) | 10 (3.9%) | 35 (9.2%) |
| Active breast feeding at enrolment | Yes | 83 (13.1%) | 34 (13.4%) | 49 (12.9%) |
| | No | 551 (86.9%) | 220 (86.6%) | 331 (87.1%) |
| Age at breastfeeding cessation (months) | Median | 5.0 | 5.0 | 5.0 |
| | IQR | 3.0–9.0 | 3.0–9.5 | 3.0–9.0 |
| Time since breastfeeding cessation (months) | Median | 19.0 | 20.0 | 19.0 |
| | IQR | 11.0–26.0 | 11.0–27.0 | 10.0–25.0 |
| Age of first solid food (months) | Median | 6.0 | 6.0 | 6.0 |
| | IQR | 5.0–6.0 | 5.0–6.0 | 5.0–6.0 |
| Time since first solid food (months) | Median | 18.0 | 18.0 | 18.0 |
| | IQR | 7.0–25.0 | 7.0–25.0 | 7.0–24.0 |
| Weight (Kg) | Median | 12.0 | 12.0 | 12.0 |
| | IQR | 10.0–14.2 | 10.1–14.0 | 10.0–14.35 |
| Height (cm) | Mean | 85.2 | 84.7 | 85.5 |
| | SD | 8.2 | 7.9 | 8.5 |
| History of prior hospital admissions | Yes | 89 (14.0%) | 27 (10.6%) | 62 (16.3%) |
| | No | 545 (86.0%) | 227 (89.4%) | 318 (83.7%) |
| Number of prior hospital admissions (if admitted) | Median | 1 | 1 | 1 |
| | IQR | 1–1 | 1–1 | 1–1 |

*(Continued)*

**Table 1.** (*Continued*)

| | | Overall (n = 634) | Seroconverters (n = 254) | Non-seroconverters (n = 380) |
|---|---|---|---|---|
| Self-reported prior episode of dysentery | Yes | 18 (2.8%) | 5 (2.0%) | 13 (3.4%) |
| | No | 616 (97.2%) | 249 (98.0%) | 367 (96.6%) |
| Prior diagnosis of being underweight or stunted | Yes | 18 (2.8%) | 5 (2.0%) | 13 (3.4%) |
| | No | 615 (97.0%) | 248 (97.6%) | 367 (96.6%) |
| | Missing | 1 (0.2%) | 1 (0.4%) | 0 (0.0%) |
| Regular use of probiotics | Yes | 215 (33.9%) | 79 (31.1%) | 136 (35.8%) |
| | No | 419 (66.1%) | 175 (68.9%) | 244 (64.2%) |
| Number of adults (age ≥15 years) in household | Median | 4 | 4 | 4 |
| | IQR | 3–6 | 3–5 | 2–6 |
| Number of children (age <15 years) in household | Median | 2 | 2 | 2 |
| | IQR | 1–2 | 1–2 | 1–2 |
| Grandparents present in household | Yes | 421 (66.4%) | 173 (68.1%) | 248 (65.3%) |
| | No | 213 (33.6%) | 81 (31.9%) | 132 (34.7%) |
| Attendance at school or kindergarten | Neither | 486 (76.7%) | 193 (76.0%) | 293 (77.1%) |
| | School | 103 (16.2%) | 44 (17.3%) | 59 (15.5%) |
| | Kindergarten | 45 (7.1%) | 17 (6.7%) | 28 (7.4%) |
| School or kindergarten hours per week, if an attendee | Median | 45 | 45 | 40 |
| | IQR | 40–48 | 40–48 | 40–48 |
| Average monthly household income (Vietnamese dong) | <1 million | 1 (0.2%) | 1 (0.4%) | 0 (0.0%) |
| | 1–3 million | 29 (4.6%) | 14 (5.5%) | 15 (4.0%) |
| | 3–5 million | 146 (23.0%) | 58 (22.8%) | 88 (23.2%) |
| | 5–10 million | 309 (48.7%) | 125 (49.2%) | 184 (48.4%) |
| | >10 million | 149 (23.5%) | 56 (22.1%) | 93 (24.5%) |
| Main household source of drinking water | Piped to residence | 423 (66.7%) | 174 (68.5%) | 249 (65.5%) |
| | Piped to public tap | 0 (0.0%) | 0 (0.0%) | 0 (0.0%) |
| | Bottled water | 204 (32.2%) | 77 (30.3) | 127 (33.4%) |
| | Well in residence | 6 (0.9%) | 3 (1.2%) | 3 (0.8%) |
| | Public well | 1 (0.2%) | 0 (0.0%) | 1 (0.3%) |
| | Rain water | 0 (0.0%) | 0 (0.0%) | 0 (0.0%) |
| | Spring | 0 (0.0%) | 0 (0.0%) | 0 (0.0%) |
| | River/Stream | 0 (0.0%) | 0 (0.0%) | 0 (0.0%) |
| | Other | 0 (0.0%) | 0 (0.0%) | 0 (0.0%) |
| Household water source for washing vegetables | Piped to residence | 603 (95.1%) | 239 (94.1%) | 364 (95.8%) |
| | Piped to public tap | 1 (0.2%) | 0 (0.0%) | 1 (0.3%) |
| | Bottled water | 0 (0.0%) | 0 (0.0%) | 0 (0.0%) |
| | Well in residence | 23 (3.6%) | 12 (4.7%) | 11 (2.9%) |
| | Public well | 4 (0.6%) | 1 (0.4%) | 3 (0.8%) |
| | Rain water | 2 (0.3%) | 1 (0.4%) | 1 (0.3%) |
| | Spring | 0 (0.0%) | 0 (0.0%) | 0 (0.0%) |
| | River/Stream | 0 (0.0%) | 0 (0.0%) | 0 (0.0%) |
| | Other | 1 (0.2%) | 1 (0.4%) | 0 (0.0%) |
| Variation in water source between wet and dry season | Yes | 17 (2.7%) | 5 (2.0%) | 12 (3.2%) |
| | No | 617 (97.3%) | 249 (98.0%) | 368 (96.8%) |
| Tendency for household water shortages | Yes | 13 (2.1%) | 5 (2.0%) | 8 (2.1%) |
| | No | 621 (97.9%) | 249 (98.0%) | 372 (97.9%) |

(*Continued*)

**Table 1.** (Continued)

| | | Overall (n = 634) | Seroconverters (n = 254) | Non-seroconverters (n = 380) |
|---|---|---|---|---|
| Water storage on the household premises | None | 249 (39.3%) | 109 (42.9%) | 140 (36.8%) |
| | Inside storage | 118 (18.6%) | 45 (17.7%) | 73 (19.2%) |
| | Outside storage | 258 (40.7%) | 100 (39.4%) | 158 (41.6%) |
| | Inside and outside storage | 9 (1.4%) | 0 (0.0%) | 9 (2.4%) |
| Coverage of water stored at the household premises | Yes | 376 (59.3%) | 141 (55.5%) | 235 (61.8%) |
| | No | 15 (2.4%) | 6 (2.4%) | 9 (2.4%) |
| | N/A or missing | 243 (38.3%) | 107 (42.1%) | 136 (35.8%) |
| Main method of water retrieval from household water store | Pouring or tap | 311 (49.1%) | 114 (44.9%) | 197 (51.8%) |
| | Scooping cup with short handle | 51 (8.0%) | 19 (7.5%) | 32 (8.4%) |
| | scooping cup with long handle | 23 (3.6%) | 10 (3.9%) | 13 (3.4%) |
| | Other | 6 (1.0%) | 4 (1.6%) | 2 (0.5%) |
| | N/A or missing | 243 (38.3%) | 107 (42.1%) | 136 (35.8%) |
| Use of toilet paper in the household | Yes | 354 (55.8%) | 147 (57.9%) | 207 (54.5%) |
| | No | 280 (44.2%) | 107 (42.1%) | 173 (45.5%) |
| Use of toilet spray-wash in the household | Yes | 603 (95.1%) | 235 (92.5%) | 368 (96.8%) |
| | No | 31 (4.9%) | 19 (7.5%) | 12 (3.2%) |
| Soap for handwashing present in the toilet room | Yes | 626 (98.7%) | 249 (98.0%) | 377 (99.2%) |
| | No | 8 (1.3%) | 5 (2.0%) | 3 (0.8%) |
| Tendency for flooding in close proximity to residence | None | 500 (78.8%) | 200 (78.7%) | 300 (79.0%) |
| | Seldom | 74 (11.7%) | 31 (12.2%) | 43 (11.3%) |
| | Moderate | 55 (8.7%) | 20 (7.9%) | 35 (9.2%) |
| | Severe | 5 (0.8%) | 3 (1.2%) | 2 (0.5%) |
| Residence in close proximity to animals | Yes | 374 (59.0%) | 150 (59.1%) | 224 (58.9%) |
| | No | 260 (41.0%) | 104 (40.9%) | 156 (41.1) |
| Type of animal in close contact | Chicken | 56 (8.8%) | 24 (9.5%) | 32 (8.4%) |
| | Duck | 2 (0.3%) | 1 (0.4%) | 1 (0.3%) |
| | Dog | 333 (52.5%) | 131 (51.6%) | 202 (53.2%) |
| | Cat | 117 (18.5%) | 43 (16.9%) | 74 (19.5%) |
| | Other | 3 (0.5%) | 0 (0.0%) | 3 (0.8%) |

seroincidence at 18 to 24 months of age did not show a statistically significant difference; 21,951 (95% CI 15,730–30,632). Comparison of the estimated seroincidence in different 12-month age periods is shown in S5 Fig and Table H of S1 Text.

## Baseline factors associated with S. sonnei seroconversion over 2 years of follow-up

Three factors were found to be significantly positively or negatively associated with the odds of observed seroconversion during the two years of follow-up in univariable logistic regression analyses. Reporting that the child had ever been breastfed was associated with increased likelihood of seroconversion (OR 2.48, 95% CI 1.20–5.09), while history of prior hospital admission (OR 0.61, 95% CI 0.38–0.99) and use of a toilet spraywash in the household (OR 0.40, 95% CI 0.19–0.85) were associated with reduced likelihood. Each of these associations remained statistically significant in a multivariable logistic regression analysis (Table 2). Notably, time since

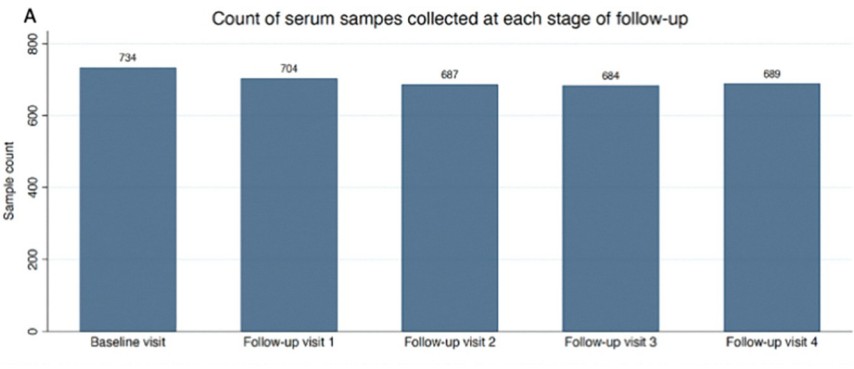

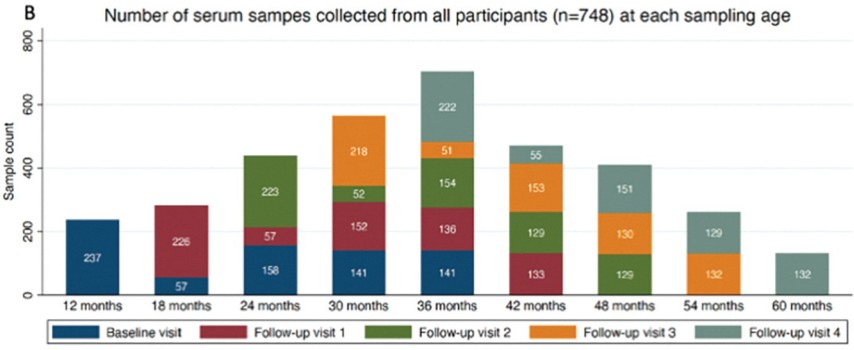

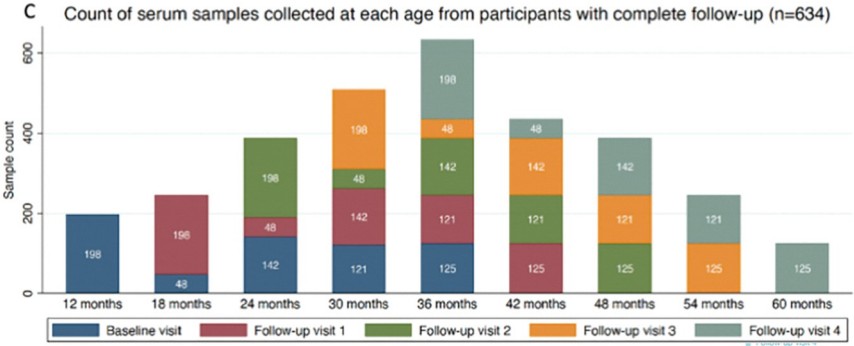

**Fig 1. Counts of serum samples collected.** Bar charts showing the numer of serum samples collected during the study period. *A*, Total number of serum samples collected at each follow-up visit. *B*, Total number of serum samples collected from participants at each age, including those taken from participants with incomplete follow-up (n = 748). *C*, Total number of serum samples collected from participants at each age, restricted to those that completed follow-up (n = 634).

breast feeding cessation at enrolment was found not to be associated with odds of observed seroconversion (OR 0.99, 95% CI 0.98–1.01).

## Discussion

*S. sonnei* is a pathogen of growing global importance as a cause of diarrhoeal illness in childhood, particularly in transitional LMIC. Until now, the modern epidemiology of *S. sonnei* as a dominant *Shigella* species in the context of rapid industrialization has been largely unexplored. Using longitudinal observation of 748 children aged <5 years in Ho Chi Minh City, we demonstrate substantial *S. sonnei* incidence in this setting, with significant reduction in infection

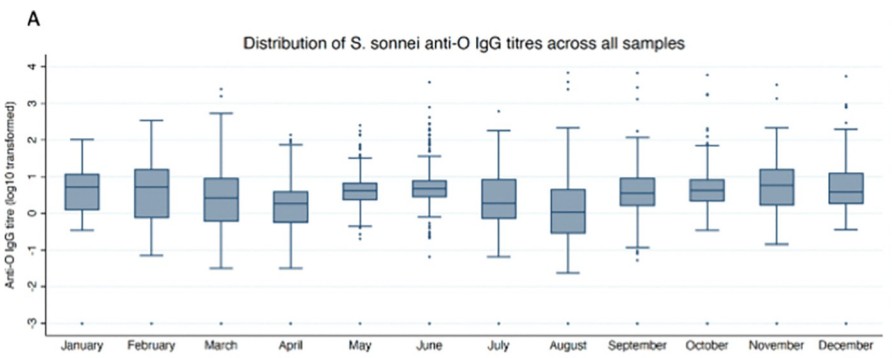

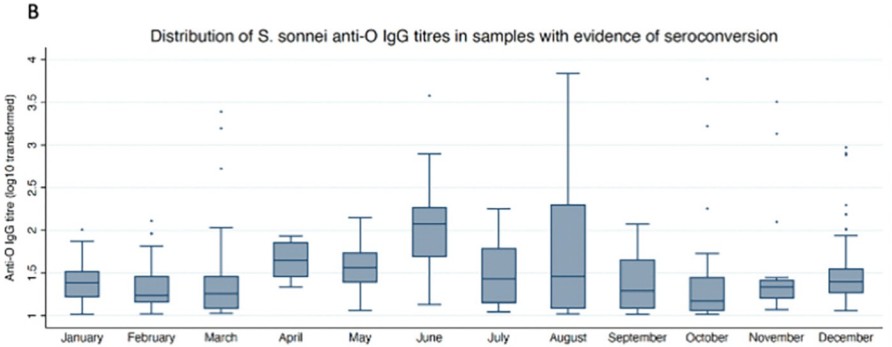

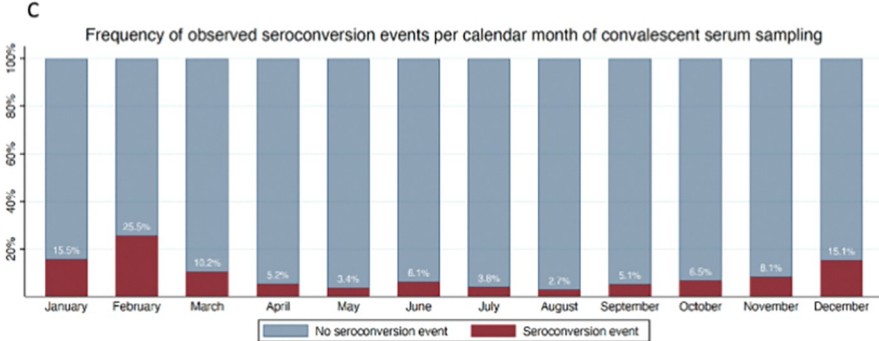

**Fig 2. Variation in *S. sonnei* anti-O IgG titres and frequency of observed seroconversion by calander month of serum sampling.** Distributions of *S. sonnei* anti-O IgG titres and the freqeuncy of observed seroconversion at each calendar month of sampling. *A*, All *S. sonnei* anti-O IgG titres (log10-transformed EU), inclusive of seronegative samples and those from participants with incomplete follow-up. *B*, *S. sonnei* anti-O IgG titres (log10-transformed EU) in convalescent samples with evidence of seroconversion only, inclusive of samples taken from participants with incomplete follow-up. *C*, Frequency of observed seroconversion events in convalescent samples, inclusive of samples taken from participants with incomplete follow-up.

incidence as age increases beyond 2 years. By virtue of serology-based surveillance, our findings capture the combined incidence of both clinically apparent and sub-clinical infections, providing valuable insight into an important reservoir of pathogen-associated antimicrobial resistance. With the development of vaccines against enteric pathogens advancing rapidly, understanding the epidemiological distribution of *S. sonnei* infection burden in transitional LMIC will be vital in ensuring optimal vaccine development and global deployment strategy.

The distribution of measured anti-O IgG levels in this cohort varied significantly by calendar month of serum sampling, as did the frequency of observed seroconversion events. This

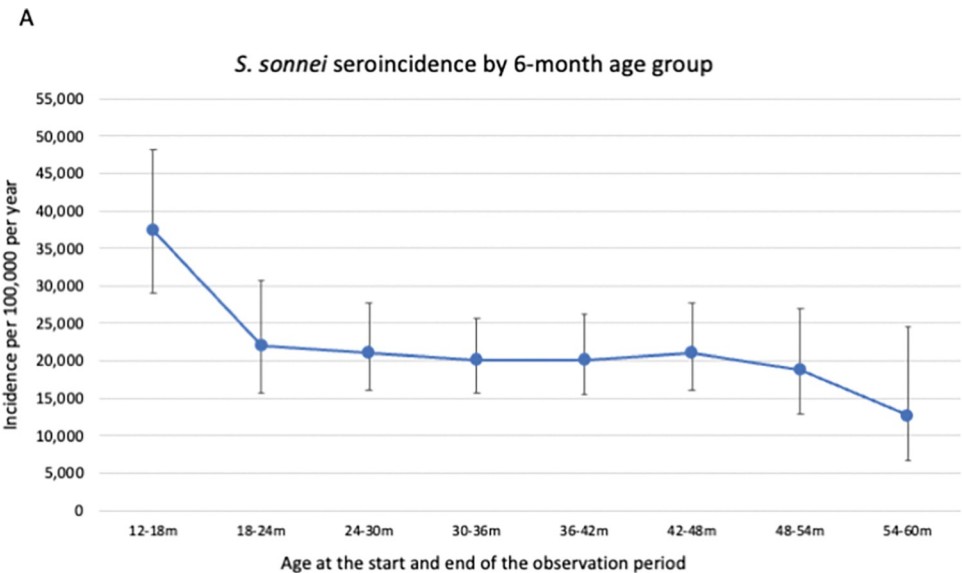

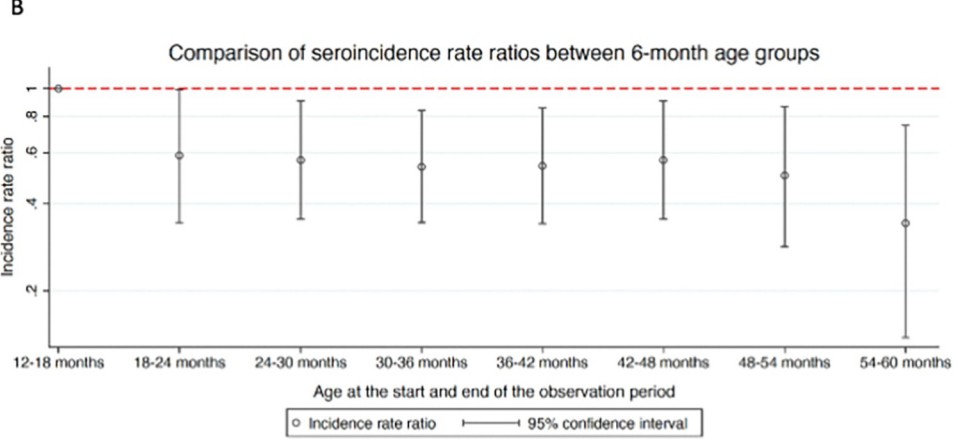

**Fig 3. Variation in *S. sonnei* seroincidence by 6-month age group.** *S. sonnei* seroincidence in different 6-month age groups, shown as: *A*, Exposures per 100,000 population, with 95% confidence intervals. *B*, Seroincidence rate ratios, with the 12–18 month age period as a reference category.

likely suggests a seasonal effect on *S. sonnei* incidence in Ho Chi Minh City, but it is difficult to draw firm conclusions on the timing of peak incidence because of the relatively large intervals between follow-up visits. Higher temperatures have been consistently identified as important climatic determinants of shigellosis incidence in Asia [17–25], and the wet season in particular coincides with peak incidence in Vietnam [17,18]. Although we observed higher titres of anti-O IgG in children of older age when considering all samples collected, incidence of seroconversion reduced significantly with age, and there was no significant variation in anti-O titres in seropositive samples, so this observation is likely incidental.

We identified three baseline factors that were independently associated with the likelihood of acquiring *S. sonnei* infection in Ho Chi Minh City; history of past or present breastfeeding at the time of enrolment, history of past hospital admission, and the use of spray-wash for toilet hygiene. Having ever breastfed was associated with significantly increased odds of observed *S.*

**Table 2. Odds ratios for observed *S. sonnei* seroconversion over 2 years of follow-up.**

| Risk factor | | Odds ratio for ≥1 observed *S. sonnei* seroconversion event over two years of follow-up (95% confidence interval) | | |
|---|---|---|---|---|
| | | Unadjusted | Age-adjusted | Multivariable model* |
| Sex | Male | 1 | 1 | N/A |
| | Female | 0.89 (0.65–1.22) | 0.89 (0.65–1.22) | N/A |
| Age (months) | 12 | 1 | N/A | 1 |
| | 18 | 1.05 (0.56–2.00) | N/A | 0.98 (0.50–1.92) |
| | 24 | 1.14 (0.74–1.77) | N/A | 1.11 (0.70–1.76) |
| | 30 | 1.07 (0.68–1.70) | N/A | 0.98 (0.60–1.58) |
| | 36 | 0.72 (0.45–1.15) | N/A | 0.74 (0.45–1.22) |
| Obesity | | 2.03 (0.88–4.69) | 2.23 (0.91–5.45) | N/A |
| Malnourishment | | 0.28 (0.03–2.38) | 0.27 (0.31–2.40) | N/A |
| Past or present breastfeeding | | **2.48 (1.20–5.09)** | **2.48 (1.20–5.10)** | **2.54 (1.22–5.26)** |
| Active breastfeeding at enrolment | | 0.93 (0.83–1.05) | 0.95 (0.84–1.07) | N/A |
| Age of breastfeeding cessation (months) | | 1.00 (0.96–1.03) | 1.00 (0.97–1.04) | N/A |
| Time since breastfeeding cessation (months) | | 0.99 (0.98–1.01) | 1.00 (0.97–1.04) | N/A |
| Age of first solid food (months) | | 1.01 (0.83–1.23) | 1.00 (0.82–1.22) | N/A |
| Time since first solid food (months) | | 0.99 (0.97–1.01) | 1.00 (0.82–1.22) | |
| Weight (kg) | | 1.00 (0.95–1.05) | 1.04 (0.96–1.12) | N/A |
| Height (cm) | | 0.99 (0.97–1.01) | 0.97 (0.93–1.02) | N/A |
| History of hospital admission | | **0.61 (0.38–0.99)** | **0.56 (0.34–0.92)** | **0.57 (0.34–0.95)** |
| Number of previous hospital admissions | | 0.58 (0.18–1.85) | 0.45 (0.14–1.49) | N/A |
| Prior diagnosis of dysentery | | 0.57 (0.20–1.61) | 0.55 (0.19–1.58) | N/A |
| Prior diagnosis of being underweight | | 0.57 (0.20–1.62) | 0.58 (0.20–1.65) | N/A |
| Probiotic use | | 0.81 (0.58–1.14) | 0.80 (0.57–1.12) | N/A |
| Number of adults (aged ≥15) in household | | 1.00 (0.93–1.08) | 1.00 (0.92–1.08) | N/A |
| Number of children (aged <15) in household | | 1.02 (0.87–1.20) | 1.03 (0.88–1.20) | N/A |
| Number of grandparents in household | | 1.14 (0.81–1.59) | 1.12 (0.80–1.57) | N/A |
| Regular attendance at school/kindergarten | Neither | 1 | 1 | N/A |
| | School | 1.13 (0.74–1.74) | 1.24 (0.78–1.96) | N/A |
| | Kindergarten | 0.92 (0.49–1.73) | 1.10 (0.55–2.22) | N/A |
| Hours per week at school or kindergarten | | 0.99 (0.93–1.05) | 0.98 (0.92–1.04) | N/A |
| Household income | >10 million | 1 | 1 | N/A |
| | 5–10 million | 1.13 (0.75–1.69) | 1.12 (0.75–1.68) | N/A |
| | 3–5 million | 1.09 (0.68–1.75) | 1.08 (0.68–1.73) | N/A |
| | 1–3 million | 1.55 (0.70–3.45) | 1.52 (0.68–3.38) | N/A |
| Main household source of non-drinking water | Piped to residence | 1 | 1 | N/A |
| | Bottled water | 0.87 (0.62–1.22) | 0.89 (0.63–1.26) | N/A |
| | Well in residence | 1.43 (0.29–7.17) | 1.46 (0.29–7.33) | N/A |
| Household water source for washing vegetables | Piped to residence | 1 | 1 | N/A |
| | Well in residence | 1.66 (0.72–3.83) | 1.67 (0.72–3.84) | N/A |
| | Public well | 0.51 (0.52–4.91) | 0.51 (0.53–4.95) | N/A |
| | Rain water | 1.52 (0.10–24.47) | 1.62 (0.10–26.14) | N/A |
| Variation in water source between seasons | | 0.62 (0.21–1.77) | 0.63 (0.22–1.80) | N/A |
| Tendency for household water shortages | | 0.93 (0.30–2.89) | 0.91 (0.29–2.81) | N/A |
| Water storage on the household premises | None | 1 | 1 | N/A |
| | Inside storage | 0.79 (0.51–1.24) | 0.79 (0.51–1.24) | N/A |
| | Outside storage | 0.81 (0.57–1.16) | 0.81 (0.57–1.17) | N/A |

*(Continued)*

**Table 2.** (Continued)

| Risk factor | | Odds ratio for ≥1 observed *S. sonnei* seroconversion event over two years of follow-up (95% confidence interval) | | |
| --- | --- | --- | --- | --- |
| | | Unadjusted | Age-adjusted | Multivariable model* |
| Coverage of water stored at the household premises | | 0.90 (0.31–2.58) | 0.88 (0.31–2.54) | N/A |
| Main method of water retrieval from household water store | Pouring or tap | 1 | 1 | N/A |
| | Scooping cup with short handle | 1.03 (0.56–1.89) | 1.05 (0.57–1.94) | N/A |
| | scooping cup with long handle | 1.33 (0.56–3.13) | 1.38 (0.58–3.25) | N/A |
| | Other | 3.46 (0.62–19.17) | 3.42 (0.62–19.00) | N/A |
| Use of toilet paper in the household | | 1.15 (0.83–1.58) | 1.13 (0.82–1.56) | N/A |
| Use of toilet spray-wash in the household | | **0.40 (0.19–0.85)** | **0.41 (0.20–0.87)** | **0.42 (0.20–0.89)** |
| Soap for handwashing present in the toilet room | | 0.40 (0.94–1.67) | 0.41 (0.10–1.72) | N/A |
| Tendency for flooding in close proximity to residence | None | 1 | 1 | N/A |
| | Seldom | 1.08 (0.66–1.77) | 1.08 (0.66–1.78) | N/A |
| | Moderate | 0.86 (0.48–1.53) | 0.87 (0.49–1.55) | N/A |
| | Severe | 2.25 (0.37–13.59) | 2.16 (0.36–13.09) | N/A |
| Residence in close proximity to animals | | 1.00 (0.73–1.39) | 1.00 (0.72–1.38) | N/A |
| Residence in close proximity to chickens | | 1.13 (0.65–1.98) | 1.14 (0.65–1.98) | N/A |
| Residence in close proximity to ducks | | 1.50 (0.09–24.06) | 1.63 (0.10–26.34) | N/A |
| Residence in close proximity to dogs | | 0.94 (0.68–1.29) | 0.94 (0.68–1.29) | N/A |
| Residence in close proximity to cats | | 0.84 (0.56–1.28) | 0.85 (0.56–1.29) | N/A |

* **Multivariable model:** Adjustment for age, calendar month of baseline sample, history of breastfeeding, history of prior hospital admission, use of toilet spraywash.

*sonnei* seroconversion, which could be explained by an increased likelihood of immune naivety in breastfed infants. Human breastmilk is rich in a number of antimicrobial factors, which serve an important role in protecting against gastrointestinal pathogens at the gut mucosa [26]. Breastfed infants may therefore have been less likely to have undergone *S. sonnei* seroconversion in the years prior to study enrolment, making them more susceptible to subsequent seroconversion during the observation period. The absence of observed association between seroconversion and the time elapsed since breastfeeding cessation does not support this hypothesis, but could be due limitations on the statistical power of this study. Regardless, it is likely that the underlying explanation for the association between baseline breastfeeding status and subsequent *S. sonnei* seroconversion is complex and multifactorial, and it is important to consider that the relationship may not be causal at all.

Conversely, history of prior hospital admission in early childhood may indicate increased likelihood of past *S. sonnei* exposure, which could explain why observed seroconversion in this study was less likely in individuals requiring hospitalization prior to enrolment. It is likely that prior exposure of the immune system to *S. sonnei* confers at least transient protection against repeat seroconversion for the majority of people. We also observed use of spray-wash for toilet hygiene to be associated with reduced likelihood of *S. sonnei* seroconversion, suggesting that such practice may be protective. This makes mechanistic sense because the use of bidet toilets has recently been shown to be effective in reducing levels of hand contamination after defaecation [27]. We did not observe a reciprocal association between use of toilet paper in the household and odds of *S. sonnei* seroconversion, but only a very small minority of individuals reported toilet paper use in the absence of spray-wash use. Further research is warranted to better characterize the role of different toilet hygiene practices in preventing gastrointestinal infection.

This study has limitations. Firstly, although the study's relatively large sample size provides confidence in the seroincidence estimates reported, the ability to ascertain risk factors for *S. sonnei* seroconversion was likely limited by inadequate statistical power for identifying associations with small effect sizes. Similarly, although the longevity of participant follow-up serves as an important study strength in enabling substantial longitudinal surveillance, the necessarily large sampling intervals make it likely for some seroconversion events to have been missed. It has been previously demonstrated that anti-O IgG responses to natural *S. sonnei* exposure peak at 2 weeks post-infection and drop to roughly two-fold higher than baseline by 10 weeks [28]. Nevertheless, this study provides an important estimate of the minimum extent of *S. sonnei* exposure in this cohort, and although the exact rate of under-ascertainment is unknown, it can be expected to have been evenly distributed across age groups. Additionally, due to the absence of available clinical data on the development of diarrheal symptoms between observation timepoints and corroborative bacterial culture results, we are only able to describe *S. sonnei* seroincidence and not the incidence of associated clinical disease. Finally, while this study provides a unique, representative insight into the minimum longitudinal exposure of infants to *S. sonnei* in this specific district of Ho Chi Minh City, the generalisability of its findings to other areas within the city and further afield remain unknown.

Our study highlights enormous exposure to *S. sonnei* within Ho Chi Minh City, with a significant reduction in the likelihood of exposure as age increases beyond 2 years. Given that this figure is likely to be substantially higher than the incidence of symptomatic disease, it suggests continued circulation of *S. sonnei* and outlines an undescribed burden of asymptomatic *Shigella* infections in an economically transitional setting. The degree to which serological responses to asymptomatic exposure give rise to subsequent protective immunity against symptomatic disease is a key consideration for future research. Our data provide valuable insight into natural IgG responses to *Shigella* O-antigen, which is likely to form an integral component of future vaccines. The approach to serological surveillance used here can be further adapted and multiplexed to enable future investigation into the impact of vaccine introductions and to identify population groups and geographical areas that should be targeted for vaccination campaigns.

## Supporting information

**S1 Fig. Variation in *S. sonnei* anti-O IgG titres by stage of follow-up in participants with evidence of multiple seroconversion events.** Scatter plots showing the variation in *S. sonnei* anti-O IgG titres by stage of follow-up for the 18 participants with completed follow-up that had evidence of >1 seroconversion event during two years of observation.
(TIF)

**S2 Fig. Variation in *S. sonnei* anti-O IgG titres by stage of follow-up in participants with consecutive four-fold rises in *S. sonnei* anti-O IgG.** Scatter plots showing the variation in *S. sonnei* anti-O IgG titres by stage of follow-up for the five participants with completed follow-up that had evidence of consecutive four-fold rises in *S. sonnei* anti-O IgG after a presumed single seroconversion event.
(TIF)

**S3 Fig. Variation in *S. sonnei* anti-O IgG titres and frequency of observed seroconversion by sex.** Distributions of *S. sonnei* anti-O IgG titres and the freqeuncy of observed seroconversion in participants of different sexes. *A*, All *S. sonnei* anti-O IgG titres (log10-transformed EU), inclusive of seronegative samples and those from participants with incomplete follow-up. *B*, *S. sonnei* anti-O IgG titres (log10-transformed EU) in convalescent samples with evidence

of seroconversion only, inclusive of samples taken from participants with incomplete follow-up. *C*, Frequency of observed seroconversion events in convalescent samples, inclusive of samples taken from participants with incomplete follow-up.
(TIF)

**S4 Fig. Variation in *S. sonnei* anti-O IgG titres and frequency of observed seroconversion by age at the time of sampling.** Distributions of *S. sonnei* anti-O IgG titres and the freqeuncy of observed seroconversion in participants of different ages at the time of sampling. *A*, All *S. sonnei* anti-O IgG titres (log10-transformed EU), inclusive of seronegative samples and those from participants with incomplete follow-up. *B*, *S. sonnei* anti-O IgG titres (log10-transformed EU) in convalescent samples with evidence of seroconversion only, inclusive of samples taken from participants with incomplete follow-up. *C*, Frequency of observed seroconversion events in convalescent samples, inclusive of samples taken from participants with incomplete follow-up.
(TIF)

**S5 Fig. Variation in *S. sonnei* seroincidence by 12-month age group.** *S. sonnei* seroincidence in different 12-month age groups, shown as: *A*, Exposures per 100,000 population, with 95% confidence intervals. *B*, Seroincidence rate ratios, with the 12–124 month age period as a reference category.
(TIF)

**S1 Text. Table A. Baseline characteristics of participants with incomplete follow-up.** Table showing the baseline characteristics of participants that did not attend all follow-up visits. **Table B. Frequency distribution of seroconversion events and variation in median anti-O IgG levels by sex.** Table showing the distribution of observed seroconversion events and median anti-O IgG levels in participants of different sexes. **Table C. Frequency distribution of seroconversion events and variation in median anti-O IgG levels by age at the time of sampling.** Table showing the distribution of observed seroconversion events and median anti-O IgG levels in participants of different ages at the time of conalescent serum sampling. **Table D. Frequency distribution of seroconversion events and variation in median anti-O IgG levels by calendar month of sampling.** Table showing the distribution of observed seroconversion events and median anti-O IgG levels in different calander months of convalescent serum sampling. **Table E. Factors associated with anti-O IgG measurements in all samples.** Table showing unadjusted and adjusted regression coefficients for anti-O IgG titres across all serum samples. Multivariable adjustment was made for sex, age at the time of serum sampling, calander month of serum sampling and calandar month of ELISA testing. **Table F. Factors associated with anti-O IgG measurements in convalescent samples at the point of observed seroconversion.** Table showing unadjusted and adjusted regression coefficients for anti-O IgG titres across all convalescent serum samples in which seroconversion was deemed to have occurred (i.e. a newly positive result). Multivariable adjustment was made for sex, age at the time of serum sampling, calander month of serum sampling and calandar month of ELISA testing. **Table G. Numbers of participants, seroconversion events and total observation times used in the seroincidence calculations for 6-month age groups.** Table showing the cumulative number of participants, number of person years of observation, and number of seroconversion events for each 6-month age group. **Table H. Numbers of participants, seroconversion events and total observation times used in the seroincidence calculations for 12-month age groups.** Table showing the cumulative number of participants, number of person years of observation, and number of seroconversion events for each 12-month age group.
(DOCX)

**S1 Data. Table of raw data.**
(XLS)

## Acknowledgments

We wish to acknowldege all the study staff, cohort members and their parents/guardians for making this study possible.

## Author Contributions

**Conceptualization:** Stephen Baker.

**Data curation:** Nick K. Jones, Phung Khanh Lam.

**Formal analysis:** Nick K. Jones.

**Funding acquisition:** Stephen Baker.

**Investigation:** Ruklanthi de Alwis, Corinne Thompson, Ha Thanh Tuyen, Tran Do Hoang Nhu, Voong Vinh Phat, Pham Duc Trung.

**Methodology:** Trang Nguyen Hoang Thu, Ruklanthi de Alwis, Corinne Thompson, Nhi Le Thi Quynh.

**Resources:** Bui Thi Thuy Tien, Hoang Thi Diem Tuyet, Lu Lan Vi, Nguyen Van Vinh Chau.

**Supervision:** Stephen Baker.

**Writing – original draft:** Nick K. Jones, Stephen Baker.

**Writing – review & editing:** Nick K. Jones, Stephen Baker.

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
