## [Decision Letter · Decision Letter 0]

17 Aug 2023

Dear Dr Jones,

Thank you very much for submitting your manuscript "The seroincidence of childhood Shigella sonnei infection in Ho Chi Minh City, Vietnam" for consideration at PLOS Neglected Tropical Diseases. As with all papers reviewed by the journal, your manuscript was reviewed by members of the editorial board and by several independent reviewers. In light of the reviews (below this email), we would like to invite the resubmission of a significantly-revised version that takes into account the reviewers' comments. 

We cannot make any decision about publication until we have seen the revised manuscript and your response to the reviewers' comments. Your revised manuscript is also likely to be sent to reviewers for further evaluation.

Sincerely,

Mathieu Picardeau

Section Editor

Mathieu Picardeau

Section Editor

Reviewer's Responses to Questions

**Key Review Criteria Required for Acceptance?**

**Methods**

-Are the objectives of the study clearly articulated with a clear testable hypothesis stated?

-Is the study design appropriate to address the stated objectives?

-Is the population clearly described and appropriate for the hypothesis being tested?

-Is the sample size sufficient to ensure adequate power to address the hypothesis being tested?

-Were correct statistical analysis used to support conclusions?

-Are there concerns about ethical or regulatory requirements being met?

Reviewer #1: (No Response)

Reviewer #2: The objectives of the study are based on logical and testable hypothesis/rationale of the study. 

The study is scientifically designed to achieve targeted objectives.

The characteristics (demographic and clinical features) of study participants are properly described.

The present study recruited substantial study subjects (3,498 samples were taken from 748 participants) which makes results more authentic and scientifically sound.

Data analysis has been done using standard statistical tools.

Ethical approval of study was taken from relevant ethical bodies (Oxford tropical research ethics committee (OxTREC) in the United Kingdom (approval 1058-13) and ethics committee of the Hospital for Tropical Diseases and Hung Vuong Hospital in Ho Chi Minh City,Vietnam).

Reviewer #3: Line 138 - Need to clarify what is meant by "sanitation" in the questionnaire. 

Line 112 - Need to add further detail about how the study will contribute to the objective of "informing the progression of S. sonnei vaccines" - currently the link is not very clear.

**Results**

-Does the analysis presented match the analysis plan?

-Are the results clearly and completely presented?

-Are the figures (Tables, Images) of sufficient quality for clarity?

Reviewer #1: (No Response)

Reviewer #2: Results are properly and scientifically presented and analysed as per the experimental strategies/methods. Good quality figures are given in the article containing information which is easily understandable and readable.

Reviewer #3: - The results should specify the characteristics of censored participants - i.e. those who didn't complete follow-up

**Conclusions**

-Are the conclusions supported by the data presented?

-Are the limitations of analysis clearly described?

-Do the authors discuss how these data can be helpful to advance our understanding of the topic under study?

-Is public health relevance addressed?

Reviewer #1: (No Response)

Reviewer #2: The conclusion supports the data presented.

The limitations of the study are absent, the author should include this section in the revised article. 

Significant finding has been discussed and compared with previous data from literature, so it will further enhance our knowledge of the topic.

Reviewer #3: - The discussion briefly mentions limitations in the statistical power of this study (line 273) - this point needs to be further elaborated, particularly as it somewhat contradicts line 291 which states that the study's strength is its large sample size. 

- Were clinical characteristic recorded for seroconverters? It would be good to quantify how the sero results compare with the detectable incidence of S. sonnei based on clinical presentations. 

- Noting that the IgG response drops to two-fold higher than baseline by 10 weeks, and the threshold used for seroconversion in the study is a four-fold increase, the 6 monthly intervals for sampling reduces the validity of the results and limits the ability of the study to meet its stated objective of measuring exposure to S. sonnei. I recognise that this is noted in the discussion as a limitation, however further information on the level of under-ascertianment is needed to increase the validity of the results.

**Editorial and Data Presentation Modifications?**

Reviewer #1: (No Response)

Reviewer #2: (No Response)

Reviewer #3: In figures 1A-1C, remove the word "simple" in the y-axis title.

**Summary and General Comments**

Reviewer #1: S. flexneri and S. sonnei have been identified as the predominant species of Shigella worldwide. S. sonnei specifically dominates in developed areas and is the second most common species after S. flexneri in economically disadvantaged populations. The proportion of S. sonnei has been progressively increasing over the years.

In this study, the ELISA method was employed, utilizing IgG antibodies against the S. sonnei O-antigen, to estimate the incidence of childhood exposure to S. sonnei in Ho Chi Minh City, where S. sonnei was the prevailing Shigella species. This study provided intriguing insights, and the information it provided holds significant importance.

However, it is well-known that serotyping methods for bacteria in the Enterobacteriaceae family, including Shigella, often encounter issues with cross-reactivity, leading to false-positive results. Therefore, to validate the conclusions drawn from the serological approach employed in this study, it is recommended to incorporate additional methods such as fluorescent quantitative PCR or bacterial isolation. These approaches can help address potential methodological errors and compare the incidence rate of Shigella infections and the rate of exposure to Shigella among populations.

Reviewer #2: In the current article (PNTD-D-23-00760), the authors highlighted the high sero-incidence of S. sonnei in Ho Chi Minh City, Vietnam. The theme and information of this article will be of great interest for readers and scientific community to know cases of Shigellosis due to S. sonnei and thus will be helpful for regional public health authorities to make treatment and prevention strategies. 

Though the study is scientifically designed and executed, but the following comments should be properly addressed.

(i)Inclusion and exclusion criteria for selection of participant may be added.

(ii) The limitation of the study should be added in the article.

(iii) The conclusion should be revised focusing on the main findings of the study and their clinical applications including future prospects.

Reviewer #3: This is a good paper and easy to read. However the 6-month sampling interval is not conducive to meeting the stated objectives of measuring sero-incidence given that IgG levels drop to two-fold above baseline within 2.5 months, and the threshold for seroconversion used in this study is a four-fold increase.

PLOS authors have the option to publish the peer review history of their article (what does this mean?). If published, this will include your full peer review and any attached files.

Reviewer #1: No

Reviewer #2: No

Reviewer #3: No
---

## [Editor Report · Decision Letter 1]

16 Oct 2023

Dear Dr. Jones,

We are pleased to inform you that your manuscript 'The seroincidence of childhood Shigella sonnei infection in Ho Chi Minh City, Vietnam' has been provisionally accepted for publication in PLOS Neglected Tropical Diseases.

Best regards,

Ana LTO Nascimento

Section Editor

Mathieu Picardeau

Section Editor

---

## [Editor Report · Acceptance letter]

24 Oct 2023

Dear Dr Jones,

We are delighted to inform you that your manuscript, "The seroincidence of childhood *Shigella sonnei* infection in Ho Chi Minh City, Vietnam," has been formally accepted for publication in PLOS Neglected Tropical Diseases.

Best regards,

Shaden Kamhawi

co-Editor-in-Chief

Paul Brindley

co-Editor-in-Chief
